# Exosomal miRNAs as Potential Diagnostic Biomarkers in Alzheimer’s Disease

**DOI:** 10.3390/ph13090243

**Published:** 2020-09-12

**Authors:** Ida Manna, Selene De Benedittis, Andrea Quattrone, Domenico Maisano, Enrico Iaccino, Aldo Quattrone

**Affiliations:** 1Institute of Molecular Bioimaging and Physiology (IBFM), National Research Council (CNR), Section of Germaneto, 88100 Catanzaro, Italy; quattrone@unicz.it; 2Department of Medical and Surgical Sciences, University “Magna Graecia,” Germaneto, 88100 Catanzaro, Italy; selene.db90@gmail.com; 3Institute of Neurology, Department of Medical and Surgical Sciences, University “Magna Graecia,” Germaneto, 88100 Catanzaro, Italy; an.quattrone@hotmail.it; 4Department of Experimental and Clinical Medicine, University “Magna Graecia” of Catanzaro, 88100 Catanzaro, Italy; maisano@unicz.it; 5Neuroscience Research Center, University Magna Graecia, 88100 Catanzaro, Italy

**Keywords:** neurodegeneration, Alzheimer’s disease, biomarkers, miRNAs, exosomes, exosomal miRNAs

## Abstract

Alzheimer’s disease (AD), a neurodegenerative disease, is linked to a variety of internal and external factors present from the early stages of the disease. There are several risk factors related to the pathogenesis of AD, among these exosomes and microRNAs (miRNAs) are of particular importance. Exosomes are nanocarriers released from many different cell types, including neuronal cells. Through the transfer of bioactive molecules, they play an important role both in the maintenance of physiological and in pathological conditions. Exosomes could be carriers of potential biomarkers useful for the assessment of disease progression and for therapeutic applications. miRNAs are small noncoding endogenous RNA sequences active in the regulation of protein expression, and alteration of miRNA expression can result in a dysregulation of key genes and pathways that contribute to disease development. Indeed, the involvement of exosomal miRNAs has been highlighted in various neurodegenerative diseases, and this opens the possibility that dysregulated exosomal miRNA profiles may influence AD disease. The advances in exosome-related biomarker detection in AD are summarized. Finally, in this review, we highlight the use of exosomal miRNAs as essential biomarkers in preclinical and clinical studies in Alzheimer’s disease, also taking a look at their potential clinical value.

## 1. Introduction

### 1.1. General Characteristics

Alzheimer’s disease (AD), the most common cause of dementia, is a neurodegenerative disorder that exhibits loss of synapses, extracellular amyloid plaques formed by the amyloid-β peptide (Aβ) and endocellular aggregates of the hyperphosphorylated tau protein.

Several studies have highlighted how alterations in energy metabolism, synaptic transmission and axon-myelin interaction may precede the appearance of neuropathological indicators. Diminished synaptic function and loss of synapses are characteristic early elements of the neuropathology of AD, usually attributed to the neuronal deposition of neurotoxic Aβ oligomers [1,2,3]. AD involves gradual and widespread loss of neurons and synapses in affected individuals, and a ~ 40% reduction in brain mass can occur at the terminal stage [4]. Many of the neurons that undergo degeneration use acetylcholine as a neurotransmitter. Therefore, the reduction of the activity of the cholinergic system is typical of the pathogenesis of AD [5]. As already mentioned, one of the two most important lesions affecting the brain of AD patients is represented by extracellular amyloid plaques. These are the result of the aggregation of Aβ molecules, which, in turn, are the product of the gradual division of the amyloid precursor protein (APP) by β-secretase (BACE1) and the γ-secretase/presenilin complex (PS1, PS2) [6]. The presence of neuronal plaques triggers an inflammatory process mediated by astrocytes and microglia with consequent induction of the immune response, which then entails the production of cytokines, interleukins and TNF-alpha, by macrophages and neutrophils, thus irreversibly damaging the neurons [7,8]. Although this primary response represents a kind of protection for the brain, it is thought that this prolonged state of chronic inflammation may be the basis of neuronal degeneration. Another major injury affecting the pathogenesis of AD is the formation of neurofibrillary tangles, which derive from the aggregation of the hyperphosphorylated forms of the tau protein; a protein that promotes the assembly and stabilization of microtubules. Its phosphorylation by some protein kinases, such as GSK-3β and Cdk-5, causes detachment from microtubules and further phosphorylation events, or the lack of dephosphorylation causes a conformational change that induces aggregation and formation of neurofibrillary tangles resulting in neuronal death [9].

### 1.2. Clinical Features and Diagnosis

Although the clinical course of AD is individual-specific, the symptoms are common to most affected individuals. Memory loss is by far the most obvious and ever-present symptom. AD is also often anticipated by mild cognitive impairment (MCI), which is followed by a progressive psychological decline in which the patient, in addition to amnesia, manifests depression, agnosia, apraxia, anomia and agraphy. MCI is an intermediate stage between age-associated cognitive impairment and dementia. It can be classified based on the underlying cognitive deficit (amnestic or nonamnestic) or the number of domains affected (single versus multiple). MCI is a heterogeneous condition with several causes such as depression, alcohol use and chronic medical diseases that need to be ruled out. Though the concept of MCI has generated controversy, it is a useful clinical construct because it helps clinicians communicate the diagnosis to patients who have cognitive impairment but do not meet the criteria for dementia. The amnestic form (aMCI), in particular, has been considered an early stage of AD [10,11]. Regarding the clinical diagnosis of AD, this includes a clinical examination, a series of laboratory evaluations, and related neuropsychological tests, in addition to an imaging examination based on the diagnostic criteria of the National Institute of Neurological and Communicative Disorders and Stroke-Alzheimer’s Disease and Related Disorders Association (NINCDS-ADRDA) [12]. However, for very early stages of the disease, as well as for a definitive diagnosis, these investigations are not yet sensitive and specific enough. Therefore, in addition to the anamnesis and neuropsychological and cognitive assessment, neuroimaging, MRI (magnetic resonance imaging) and PET (positron emission tomography) must be carried out. Even if Aβ neuroimaging using PET, and research of cerebrospinal fluid biomarkers, present good diagnostic accuracy for AD, the high costs and invasiveness of these methods, respectively, do not allow easy application [13]. Finally, only an autopsy can confirm a diagnosis of AD. As for the mechanisms that underlie the development of AD, the greatest risk of disease is linked to genetic factors, especially in family forms. In recent years, knowledge of the genetic variants that contribute to Aβ processing has evolved considerably [14]. It started from the discovery of various mutations in genes such as *APP*, *PSEN1*, *PSEN2* which are related to the familiar form of AD, and polymorphisms in the *APOE* gene, as risk factors for early-onset and late-onset forms [15]. Recently, using genome-wide association studies, several additional genetic risk loci have been identified with regard to genetically complex forms of AD [16]. As reported in a recent review [17], many other risk factors are associated with AD, such as age, familial inheritance, vascular factors, the immune system, exposure to infectious agents and epigenetic factors. Epigenetic mechanisms at the interface of genetic and environmental risk factors may contribute to the onset and progression of AD. Recent studies suggest that at the basis of neurodegenerative diseases there is an involvement of epigenetic mechanisms and, therefore, of an altered post-transcriptional regulation [18,19,20]. These observations indicate that neurodegenerative processes could be the result of the alteration of different cellular pathways in which miRNAs can play a significant role as a unifying element of very heterogeneous phenotypes [21,22]. Given the complexity and heterogeneity that characterize AD, its molecular bases are not still completely clear. However, despite extensive research on AD, we do not yet have drugs or agents capable of preventing the disease and/or limiting its progression, and we do not know specific biomarkers for the early diagnosis of AD. Consequently, there is an absolute need to have specific biomarkers for the early diagnosis of AD with the aim of implementing early treatment designs. To date, the search for new biomarkers is, therefore, very active. In this context, miRNAs offer an opportunity for biomarker research.

## 2. miRNAs: Biogenesis and Function

miRNAs are important regulators of gene expression. Therefore, alteration of miRNA expression can result in dysregulation of key genes and pathways that contribute to disease development [23]. miRNA dysregulation has been shown across multiple pathologies, including central nervous system diseases [24]. miRNAs represent a group of small, single-strand endogenous noncoding RNAs between 21 and 25 nucleotides in length, resulting from the processing of much longer primary transcripts. The process by which mature miRNAs are formed is complex and consists of several steps. It begins in the nucleus and is completed at the level of the cytoplasm where the pairing of the mature miRNA with the target messenger takes place. miRNAs repress gene expression by binding to complementary sequences in the 3′ untranslated region (3′UTR) of mRNAs to target them for degradation and thereby prevent their translation. From a functional point of view, it has been shown that a single miRNA can prevent the translation of numerous messenger RNAs by altering the specific recognition of the seed sequence, which is 2–8 nucleotide from the 5′end and, based on the degree of nucleotide complementarity, will increase the inhibition of translation and/or degradation of their targets that are often involved in a functional interacting pathway [25]. In particular, if the seed sequence is perfectly complementary to the 3’UTR region of mRNA, the latter will undergo degradation. If, on the contrary, the complementarity is imperfect, the translation process will be blocked [26,27] (Figure 1).

miRBase is the main public archive online for miRNA sequences and annotations (MirBase; http://www.mirbase.org/) [29]. The latest release of the miRBase database (v22) contains about 3000 mature sequences, and it has been estimated that they are able to regulate over 60% of protein-coding genes [30]. Since their ubiquitous function in gene regulation, they take part in all cellular processes, such as homeostasis, proliferation, differentiation, apoptosis and response to stress, and in more complex processes such as embryonic development and growth [31]. An alteration of miRNA expression appears to be associated with pathophysiological changes, and numerous studies have been carried out on alterations of miRNA expression in many diseases. In particular, many works concern the role of miRNAs in brain development, in the processes of normal brain aging and in various neurological disorders [32]. Recent evidence suggests a potential involvement of miRNAs in various neurological disorders and neurodegeneration, confirming their importance in CNS functions and pathologies. [33,34].

In addition to gene regulation, miRNAs play a key role in intercellular signaling. Most miRNAs are found inside the cell, but they can migrate outside and, as circulating miRNAs, they can be found in biofluids [35,36]. They are released in blood, and other fluids, through tissue damage, apoptosis and necrosis [37], or through active passage in microvesicles, exosomes, or through binding to a protein [38,39].

Circulating miRNAs are very stable in conditions that would normally damage most RNAs, such as high or low temperature, pH modification and freeze/thaw cycles. The stability of circulating miRNAs is due both to their association with various RNA-binding proteins or lipoprotein complexes and to their inclusion in microparticles. Circulating miRNAs are known to be both contained in exosomes and in complexes with proteins such as argonaute 2 (Ago2) [40]. This type of packaging is crucial to avoid the degradation of miRNA by the RNAses present in biofluids [41]. Lately, it has been proposed that exosomes could transfer miRNA to regulate biological activities in recipient cells [42,43], indicating that variations in miRNA release could be an intercellular communication mechanism. Since the miRNAs are relatively small molecules, do not undergo postprocessing modifications and have a less complex chemical structure, they have numerous advantages over conventional protein-based markers. Because of both their ability to reflect cellular status and the ability to measure them in biofluids, exosomal and free-circulating miRNAs result as new classes of disease biomarkers. Currently in the literature, it is possible to find numerous studies that have focused their attention on the possibility of using circulating miRNAs as markers of neurodegeneration, considering their altered expression [44,45,46]. Research on the association of exosomal miRNAs with AD could lead to the discovery of diagnostic and prognostic biomarkers and increase understanding of the biological mechanisms underlying the disease. In this review, we illustrate the existing knowledge on circulating miRNAs, in particular exosomal miRNAs, and their involvement in AD, highlighting their potential use as diagnostic biomarkers in pathology.

## 3. miRNAs and Alzheimer’s Disease

Approximately 70% of experimentally detectable miRNAs are expressed in the human nervous system where they ensure the complex development and function of this organ [47]. Due to the involvement of miRNAs in all the fundamental processes of the system nervous, from development and differentiation to the functionality of mature neurons, various groups in recent years have focused on investigating a possible involvement of miRNA in brain development and neurodegeneration [48]. In particular, impaired regulation of miRNAs has been linked to neurodegeneration, and deregulation of particular miRNA profiles has been reported in different pathologies like AD, Parkinson’s disease, amyotrophic lateral sclerosis, multiple sclerosis and epilepsy [28,49,50,51]. The data presented in the work of Burgos et al., one of the largest data sets, concerning the miRNAs in cell-free body fluids in neurodegenerative disease like AD and PD, was the first to use next-generation deep sequencing (NGS) to compare the profiles from CSF and blood serum [52]. Using next generation small RNA sequencing, Burgos and his colleagues assessed a complete check of the miRNAs detected in CSF and blood. In AD patients, compared with controls, a total of 41 and 20 miRNAs were determined to have different expression levels in CSF and serum, respectively. Moreover, the data collected indicate that extracellular miRNAs are an expression of cellular changes in disease and can be used to evaluate AD progression as well as response to drug therapy, but validation of these miRNAs in larger patient cohorts will enable the discovery of crucial miRNA biomarkers that are most associated with certain neurodegenerative diseases and, therefore, with the severity of the latter [52].

As mentioned above, AD represents the most common form of dementia and, for this reason, in recent years the research groups that have focused on investigating a possible involvement of miRNAs in AD have been numerous [53]. However, there is little knowledge about the expression of miRNAs in patients’ brains and, in general, the molecular networks to which they are connected. The studies conducted by Sheinerman et al. [54] support the hypothesis that the degeneration of neurons and synapses, which is the basis of the pathological processes of neurodegenerative diseases, and even before MCI, can be identified in vitro through the quantitative analysis of miRNA. miRNAs play a key role in the four fundamental processes in the onset of AD: Aβ accumulation, tau-dependent toxicity, inflammation and neuronal death [55]. There are several miRNAs whose expression is altered in Alzheimer’s patients [56]; for example, there is the group of miR-15/107 which, among various functions, has that of regulating the *APP* gene responsible for the production of APP and BACE1, which encode an enzyme involved in the transformation of the APP protein in the β-amyloid end product. Furthermore, other miRNAs control the expression of proteins implicated in the APP elaboration pathway, such as ADAM10/α-secretase [57,58,59], and these studies suggest that many miRNAs are involved in AD pathogenesis through both APP processing and tau regulation.

Lau et al., presented an overview of the altered miRNAs in AD, established by profiling the hippocampus and the prefrontal cortex of a cohort of patients with late-onset AD (LOAD) and 23 controls. They found deregulation of a small subset of miRNAs in both brain areas of AD patients. Among these, the most dysregulated during the disease phases was miR-132-3p. This study represents the first large-scale analysis of AD dysregulated miRNAs. Indeed, as the disease progresses, neurons present a lower amount of miR-132-3p and an accumulation of hyperphosphorylated tau, indicating the downregulation of miR-132-3p as a new disease biomarker [60].

Some members of the Let-7 family (let-7f, let-7b and let-7i), as well as miR-9, miR-181 and miR-29, appear to be involved in inflammation and immunological responses, processes that precede the onset of AD [43]. The alteration of miRNAs in the CNS can affect many cellular functions, which are not limited solely to the etiology of AD. The detection of the same miRNAs in different neurological disorders can indicate common pathways of neuronal dysfunction and degeneration. For example, the overexpression of miR-181b in schizophrenia, or let-7 and miR-125b in Down syndrome have also been observed in sporadic AD patients, which can indicate the same disease mechanisms [61].

## 4. Exosomes

### 4.1. Exosomes Biogenesis

Many cell types release a class of nanosized vesicles named exosomes, with diameters ranging from 30 to 150 nm [62], into the extracellular microenvironment, and are involved in intercellular communication. Exosome biogenesis begins with the endosome system. During endocytosis, the vesicles that are formed at the level of the plasma membrane merge to form early endosomes. These mature into late endosomes and from them into multivesicular bodies (MVBs) that merge directly with the plasma membrane releasing the exosomes into the extracellular space. The biogenesis of exosomes and the sorting of cargo proteins, and their release, require the endosomal ordering complex necessary for transport (ESCRT) complex. In many biofluids there are exosomes which contain various molecules like proteins, lipids, and both mRNA and noncoding RNAs, which mirror the phenotypic state originating them [63]. In addition, exosomes present several proteins common to all cytotypes, such as CD63, CD81, CD9, TSG-101, ALIX, and HSP70. These last represent exosomal markers and are used for this purpose to identify their presence [64].

### 4.2. Exosomes: Biological Functions

Different types of cells like neurons, adipocytes and fibroblasts are able to release exosomes and, therefore, they can be found in many biofluids such as blood, urine, saliva and amniotic fluid [65]. Exosomes can be purified from biological fluids and cell cultures in vitro using a variety of strategies. Several papers offer an overview of the most used techniques for the isolation of exosomes from biological fluids and for the extraction of their cargos, such as miRNAs. [66,67,68,69]. Exosomes have been compared to cellular garbage bags, and so can expel excess and/or nonfunctional cellular components [70]. The biological and functional characteristics of exosomes are linked to the cell of origin and also to the state of the tissue from which they originate. Exosomes have a key role in many biological processes such as inflammation, apoptosis and intercellular signaling, due to their capability to transfer RNA, proteins, enzymes and lipids. This is reflected in different physiological and pathological processes in various diseases, including neurodegenerative diseases [71].

### 4.3. Exosomes: New Molecular Targets of Diseases

In neurodegenerative diseases, exosomes, due to their role as transporters of biomolecules, have aroused considerable interest [72]. In this regard, many studies highlight how exosomal content, which may be involved in neurological damage, may offer a future approach for the diagnosis of neurodegenerative diseases [58,73,74]. In CNS, not only neurons and astrocytes secrete exosomes into the extracellular environment, but also microglia and oligodendrocytes. Exosomes are found associated with neurodegenerative diseases such as Parkinson’s disease, AD, Huntington’s disease and prion diseases, which are characterized by progressive neuronal degeneration and often associated with misfolded proteins [75]. Exosomes provide an important cell-cell communication mode by delivering their load to target cells, thus contributing to disease progression. Infectious prion particles present in exosomes have been shown to propagate within cells as well as poorly folded proteins, such as amyloid-β and tau in AD and α-synuclein in PD, and they exploit exosomes for their diffusion. Indeed, due to the exchange of proteins and genetic materials, exosomes participate in physiological processes including cell growth, immune regulation, angiogenesis, neuronal communication and cell migration [76], and in the pathogenesis of various diseases such as AD. It has been observed that exosomes induce apoptosis, and consequently neuronal loss, by mediating the diffusion between cells of Aβ protein and hyperphosphorylated tau [77]. On the other hand, through microglial uptake, exosomes may be beneficial, as they could induce degradation and clearance of brain amyloid-beta. Finally, the discovery that exosomes also contain messenger RNAs and miRNAs shows that they could be carriers of genetic information capable of modulating the expression of target genes in receiving cells. Due to these characteristics, exosomes are very interesting in the development of new therapeutic approaches [78]. The ease of detection of miRNA in exosomes isolated from biological fluids is one of the characteristics that makes these molecules new potential noninvasive disease biomarkers. According to a recent review of the literature, exosomal miRNAs would seem to be better sources of biomarkers than other circulating miRNAs, by virtue of their quantity, stability and quality [79].

## 5. The Role of Exosomal miRNAs as Potential Biomarkers in Alzheimer’s Disease

It is known that nerve cells synthesize and release exosomes, which cross the blood-brain barrier (BBB) and are thus detected in the blood and peripheral fluids [80]. As previously mentioned, exosomes are found in all biological fluids including blood and blood components, serum and plasma. This makes exosomes excellent biomarkers that reflect the pathological state of neurodegenerative diseases, including AD. Exosomes are considered as fingerprints, or signatures, of the donor cell due to their specific profile of biomolecule content that can reflect the cellular origin and its physiological state [63]. Indeed, change in exosomal content can represent an early index for both diagnosis and treatment of the disease [81]. It is well known that miRNAs regulate AD-associated proteins in the brain and are stable in CSF and blood as they transport through biofluids within exosomes [82]. The investigation of exosomal miRNA from serum samples from patients and healthy subjects is a noninvasive and feasible approach to determine biomarkers for AD, also because the miRNAs encapsulated in the exosomes are free of endogenous RNA contaminants, e.g., ribosomal RNA.

### 5.1. Exosomal miRNAs in CSF

Concerning exosomal miRNAs in CSF, the experience with AD is still very limited.

Gui and colleagues first observed the presence of miRNAs in CSF exosomes of PD and AD patients, as well as revealing substantial abundance in CSF exosomes and mRNA transcripts and long noncoding RNAs. In order to evaluate the exosomal miRNA deregulation in the CSF of patients with PD and AD, they analyzed the expression of 746 miRNAs using the TaqMan miRNA array. Authors observed a significantly deregulation of miR-29c, miR-136-3p, miR-16-2, miR-331-5p, miR-132-5p and miR-485-5p in AD patients compared to healthy controls [74].

Riancho and collaborators, in a recent study, compared the miRNA levels in whole CSF and in the CSF exosome-enriched fraction in AD patients and healthy controls, in order to examine a profile of miRNAs previously reported by other authors. A panel of 760 miRNAs was used to examine miRNAs isolated from the whole CSF of AD patients and healthy controls, and 14 differentially expressed miRNAs were identified. Further, RT-qPCR analysis showed the presence of miR-9-5p in 50% and miR-598 in 75% of whole CSF control samples, while they were not detected in any AD CSF sample, but a surprisingly marked inversion in the miRNA exosome- enriched fraction was observed. Indeed, miR-9-5p was found in seven out of 10, and miR-598 in about 80% of exosome-enriched CSF samples from AD, compared to AD whole CSF samples [83]. These results might suggest increased exosome trafficking in AD, in line with the tendency observed by other authors [84,85].

In a more recent study, McKeever and colleagues evaluated the expression of exosomal miRNAs in the CSF of sporadic patients with early onset of disease, observing a decrease in miR-16-5p, miR-451a and miR -605-5p, and an increase in miR -125b-5p, in patients compared to healthy controls. In addition, the study showed that in a cohort of late-onset AD patients, miR-451a and miR-605-5p were both reduced, as opposed to miR-125b-5p, and there were no significant differences in miR-16-5p compared to healthy controls. Hence, these results showed an altered expression of 3 exosomal miRNAs derived from CSF in patients with early and late-onset disease [86]. So, given the exosomal miR-16-5p deregulation only in the early-onset group, this mRNA could represent a specific biomarker linked to pathological mechanisms related to the early onset of disease. Overall, although these studies led to the identification of a differential expression profile of previously identified and new miRNAs, due to the difficulty, invasiveness and risk of collecting CSF it is not suitable for routine screening and testing.

### 5.2. Exosomal miRNAs in Blood Components

Due to the limitations related to CSF, attention turned to the identification of blood-based biomarkers. In a 2014 study, Cheng et al. used NGS to profile miRNA, to identify differences in profiles within peripheral blood compared to cell-free plasma or serum and exosomes. miRNAs have been found enriched in exosomes; in particular the miR-451a, miR-223-3p, miR-16-5p, miR- 191-5p, miR-486-5p, miR-126-3p, miR- 484, miR-126-5p, miR-26a-5p and let-7b-5p showed more reads compared to miRNAs detected in cell-free samples. From this it emerges that exosomal miRNAs protected by the action of RNAase represent a reliable resource as disease biomarkers [87].

Lui et al. suggested that miR-193b may be associated with the development of AD and, in particular, exosomal mir-193b could represent a noninvasive biomarker of MCI and dementia of Alzheimer-type (DAT). Patients with MCI and DAT had lower exosomal miR-193b expression compared to controls. Interestingly, as pointed out by the authors, DAT patients had lower exosomal miR-193b levels in blood compared with the MCI group. Moreover, a lower exosomal expression of miR-193b was also observed in CSF of patients with DAT [58].

In two different works involving the deep sequencing of the blood components serum and plasma, exosomal miRNA analysis led to the identification of differentially expressed miRNA panels in AD patients, suggesting that an exosomal profile of miRNAs could represent an effective screening tool for AD disease.

In a study of sequencing analysis, Cheng and collaborators analyzed exosomal serum miRNAs, in order to identify different expression of these in AD, validating them by qRT-PCR. An initial screening of exosomal miRNAs revealed 14 upregulated and three downregulated miRNAs (see Table 1) which can, therefore, be considered as biomarkers [88].

In the work of Lugli et al., the profile of 20 miRNAs (see Table 1) was significantly different in AD patients. Between these, seven miRNAs (miR-342-3p, miR-141-3p, miR-342-5p, miR-23b-3p, miR-24-3p, miR-125b-5p, miR-152-3p) were highly informative in a machine-learning model for predicting AD status. In particular, the expression of the brain-enriched miR-342-3p, in the patients’ group, was statistically significant (*p* = 0.0007) [89].

The study of Wei et al. concerned the analysis, in AD patients, of the serum and exosomal expression of miRNAs related to neuroinflammation: miR-137, miR-155 and miR-223. In this regard, they considered a cohort of 11 AD patients at their first clinic visit, 11 AD patients on drug treatment, 10 patients with vascular dementia and 16 age-matched nondementia controls. Their results showed a reduced expression of serum exosomal miR-223 in the dementia group compared to the healthy subjects (*p* < 0.01), while the same miRNA was more expressed in the group of patients with vascular dementia than in the AD group (*p* < 0.05). Moreover, an interesting finding was that the serum exosomal miR-223 was higher in the AD patients already under drug therapy than in the AD first-visit patients (*p* < 0.05). The study identified the serum exosomal miR-223 as a promising biomarker to diagnose dementia and assess disease progression [91].

Yang et al., in order to explore the potential value of miR-135a, miR-193b and miR-384 as biomarkers for AD diagnosis, measured their expression levels in the serum from MCI, dementia of Alzheimer-type, Parkinson’s disease with dementia and vascular dementia patients. In the AD patients compared to normal controls, serum exosomal miRNA: miR-135a and miR-384 were up-regulated as opposed to miR-193b. Exosomal miR-384 was the best miRNA to discriminate in differential diagnosis, and the combination of miR-135a, miR-193b and miR-384 showed a better response in predicting the early diagnosis of AD. In addition, the study highlighted the upregulation of miR-135a and miR-384 levels in the MCI and AD groups compared to the control group, and the significant downregulation of the exosomal miR-193b in the MCI and AD groups [92].

Finally, in the work carried out by Gámez-Valero et al. they analyzed the miRNA profile associated with plasma-derived extracellular vesicles (EVs) (such as exosomes) from subjects with dementia with Lewy bodies (DLB), AD and healthy controls. The results indicated that specific EV-associated miRNAs were dysregulated between patients with DLB and AD, and this could be a valuable aid in differential diagnosis. In particular, from this study emerged two miRNAs differentially downregulated in AD patients compared to DLB and aged-matched controls (miR-451a and miR-21-5p), and four miRNAs were also significantly down-regulated in AD patients compared to the control cohort (miR-23a-3p, miR-126-3p, let-7i-5p and miR-151a-3p). Although this work concerned a limited study cohort, it highlighted the possibility of discriminating between the two most common forms of degenerative dementia [90].

The studies and their findings presented in this review focused on particular miRNAs believed to be potential biomarkers for AD. As summarized in Table 1, the studies used real-time microarray and qRT-PCR methods on human biofluid samples such as serum, plasma and CSF from patients with dementia due to MCI and AD, and in healthy controls, to quantify the expression of exosomal miRNAs.

miRNAs are stable in biological fluids, including serum, plasma and CSF, as they travel through these within exosomes and can be readily detected using standard molecular biology techniques such as quantitative PCR (qPCR). However, from a critical analysis of the above works, the main limitation may be the lack of overlap in the miRNAs detected in the different studies. This could be due to the fact that the variables in these studies are many, such as the nature of the biological liquid (serum/plasma vs. CSF), the methods for exosome isolation, the methods of miRNAs quantification (such as microarrays and RNA sequencing) and data analysis. Therefore, we think that the need for very standardized parameters is fundamental. Among these the most important are patient recruitment, gender, inclusion and exclusion criteria, drugs, the most appropriate validation methods, normalization and statistical data analysis. However, more studies are needed to state whether exosomal mRNAs can function as biomarkers of AD. This field of investigation is constantly evolving, research is still ongoing and many other miRNAs appear to be useful as biomarkers of AD.

The fact that patients with AD are characterized by alterations of miRNAs in biological fluids, and many of their target genes such as presenilins, BACE-1, APP are directly involved in the pathophysiology of AD, generates great interest in the research of the mechanisms involved in the regulation of target genes belonging to the main pathways affected by AD, such as neuroinflammation, neurodegeneration, neurogenesis, oxidative response and neuronal plasticity.

## 6. Discussion

The increase in diagnostic tests and preventive therapies for neurodegenerative pathology is difficult given the complexity of the pathological mechanisms that underlie them, and also the trouble of having an accurate diagnosis from the beginning of the disease. Currently, the diagnosis of AD is established on the basis of the patient’s cognitive functions, and through imaging techniques and biochemical analyzes. It is well known that CSF and PET markers of amyloid- β and tau proteins are accurate in identification of neuropathological changes of AD, but their employ as biomarkers is limited by invasiveness and /or high costs. Considerable attempts have been made for the purpose of developing both an effective therapy and a diagnosis aimed at identifying AD before neurological damage becomes irreversible. A line of research, which appears to be able to give results, identifies miRNAs as possible candidates. miRNAs are molecules that play important biomolecular roles within cellular networks and, given their small size, are found in biological fluids such as circulating miRNA. In this review, we have summarized the roles of exosomal miRNAs as biomarkers and pathological mediators. Exosomes are involved in the aberrant pathological processes of neurodegenerative disorders, and the transfer of miRNA mediated by exosomes has been identified as a new process of genetic exchange among cells [93]. Several features make exosomal miRNAs useful biomarkers of central nervous system diseases, including AD. These characteristics include:(1)Exosomal miRNAs are easily accessible and exosomes are abundantly present in various biological fluids and can be extracted with relative ease without excessively invasive procedures [66,94]. Indeed, exosomes are able to cross the BBB and, therefore, can be detected through simple venous sampling.(2)Exosomal miRNAs are less subject to the process of degradation than free miRNAs [87,93] because they are protected by the RNases found in biological fluids. This peculiarity of exosomal miRNA allows identification of temporal changes in their expression during the course of disease, and allows mediation of cell signaling related to the disease in a more lasting way.(3)Usually exosomal miRNAs have different expression patterns compared to cellular or free miRNAs [83,87]. Indeed, the miRNAs are packaged within the exosomes through specific mechanisms, although these specific mechanisms are not yet fully identified [95].(4)Exosomes are strongly enriched with miRNA, unlike the cells of origin and blood without cells. Thus, exosomes derived from body fluids, and the miRNAs contained in them, were studied for biomarker profiling [87].(5)Exosomal miRNAs from CNS can provide information from their originating cells in order to accurately follow the state of the nervous system cells and brain tissues [96].

In addition, considering the peculiar physiological and biological characteristics of exosomes, they could represent a valid aid in therapy. Indeed, exosomes can help exchanges between peripheral circulation and the CNS crossing the BBB. Moreover, they can be engineered using specific ligands on their surface in order to provide them targeting capability [97]. Based on this, they can be used as nanovesicles delivering siRNAs/miRNAs to the CNS [98]. The efficacy of treatment through siRNA delivery was first demonstrated in animal models by Alvarez-Erviti et al. The administration of rabies viral glycoprotein (RVG)-targeted exosomes loaded with exogenous siRNA to AD mice, demonstrated a significant reduction of Aβ deposits in the brain of animals [99]. Therefore, for gene transfer and targeting of therapeutic molecules on certain cell types, the release of exosomes is of fundamental importance. Hence, exosomal miRNAs could be useful for the development of new therapeutic strategies, employing them as drugs for CNS delivery in order to regulate the expression of genetic diseases [24,87]. Together, these studies define exosomes as potential therapeutic vehicles for the treatment of AD, and that exosomal miRNAs can play a central role as biomarkers in preclinical and clinical studies.

## 7. Conclusions and Future Perspectives

The presence in peripheral body fluids, such as CSF and serum, of molecules that could act as biomarkers for the diagnosis of neurodegenerative diseases has become an active area of research. As CSF is directly in contact with the extracellular space of the brain and can mirror the biochemical alterations that occur in the brain, it is an optimal source of AD biomarkers, and circulating miRNAs are attractive candidates for monitoring disease. Regarding the choice of the optimal biological fluid to be used for analysis, the use of serum, instead of CSF, provides appreciable results and circulating miRNAs are interesting candidates for disease monitoring. In the progression of AD, the exosomes are relevant agents, as they can carry small genetic fragments and toxic proteins between cells and extracellular fluids.

Although more standardization is needed in miRNA analysis protocols, specific profiles of exosomal miRNAs as circulating biomarkers, as well as other specific clinical biomarkers and instrumental tests, could not only improve early diagnosis of AD but could also provide new insights into disease screening and prevention. An miRNA panel could be used for specific functions, such as disease stage, prediction of the risk of conversion from MCI to AD and disease progression. However, more detailed studies of miRNA expression in AD and AD-type dementia are needed to understand miRNAs at different stages of disease progression. In addition, a more uniform methodological process for the screening of peripheral biofluids is urgently needed in order to identify and establish circulatory miRNAs as peripheral biomarkers. We think this review may support a more critical consideration of the clinical utility of exosomal miRNAs.

## Figures and Tables

**Figure 1 pharmaceuticals-13-00243-f001:**
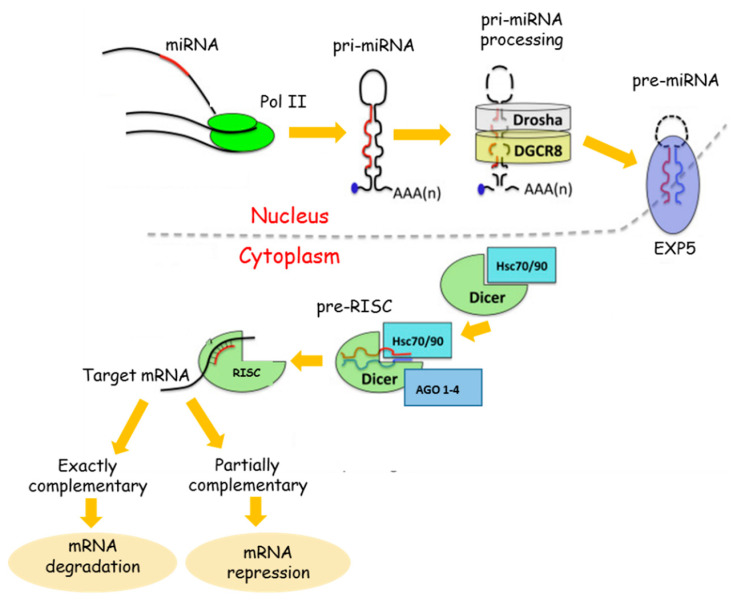
miRNA Biogenesis. The long primary miRNA (pri-miRNA) with its hairpin structure is processed by the Drosha-DCGR8 complex. The microprocessor complex, formed by Drosha and its cofactor DCGR8, leads to the maturation of the pri-miR into precursor miRNA (pre-miRNA). After Drosha processing, pre-miRNAs are exported into the cytoplasm by Exportin 5 (EXP). In the cytoplasm, another RNase guides the maturation of pre-miRNA into the duplex form, i.e., Dicer. Dicer interacts with the double-strand hairpin structure of the cytosolic pre-miRNA and, in collaboration with RNA-binding protein, cleaves the RNA duplex. Further, the mature miRNA, the red strand, is loaded directly into the RNA-induced silencing complex (RISC), whereas the passenger miRNA, the blue strand, is usually degraded. The RISC complex needs the help of two heat shock cognate proteins (HSC70 and 90) that mediate the opening of the argonaute (AGO) protein to receive the miRNA duplex. The RISC receives the duplex, unwinds it, and degrades the passenger miRNA. The pairing between the RISC-mature miRNA and the seed sequence on the target mRNA determines mRNA degradation or mRNA repression. Adapted from Cava et al. [28].

**Table 1 pharmaceuticals-13-00243-t001:** Potential exosomal miRNAs biomarkers for Alzheimer’s disease clinical diagnosis.

Source	Upregulated miRNA	Downregulated miRNA	Study Population	Exosomes Isolation Methods	miRNAs Detection Methods	Ref.
CSF	miR-132-5p miR-485-5p	miR-16-2 miR-29c miR-136-3p miR-331-5p	47 PD 28 AD 27 HC	Ultracentrifugation	microarray analysis	[74]
CSF		miR-9-5p miR-598	10 AD 10 HC	Commercial isolation kit	microRNA panel qRT-PCR	[83]
CSF	miR-125b-5p	miR-16-5p miR-451a miR-605-5p	LOAD 13 YOAD 17 HC 12	Commercial isolation kit	microarray analysis qRT PCR	[86]
CSF and Blood		miR-193b	51 DAT 43 MCI 84 HC	Commercial isolation kit	qRT-PCR	[58]
Plasma		miR-23b-3p miR-141-3p miR-185-5p miR-342-3p miR-342-5p miR-338-3p miR-3613-3p	46 AD 41 HC	Ultracentrifugation	NGS	[89]
Plasma		let-7i-5p miR-21-5p miR-23a-3p miR-126-3p miR-151a-3p miR- 451a	10 AD 18 DLB 15 HC	Size Exclusion Chromatography	NGS qRT-PCR	[90]
Serum	miR-15a-5p miR-18b-5p miR-20a-5p miR-30e-5p miR-93-5p miR-101-3p miR-106a-5p miR-106b-5p miR-143-3p miR-335-5p miR-361-5p miR-424-5p miR-582-5p miR-3065-5p	miR-15b-3p miR-342-3p miR-1306-5p	AD 23 MCI 3 HC 23	Commercial isolation kit	NGS qRT-PCR	[88]
Serum		miR-223	22 AD 10 VAD 16 HC	Commercial isolation kit	qRT-PCR	[91]
Serum	miR-135a miR-384	miR-193b	107 DAT 101 MCI 30 PD 20 VaD	Commercial isolation kit	qRT-PCR	[92]

CSF: Cerebrospinal fluid; PD: Parkinson’s disease; AD: Alzheimer’s disease; HC: healthy control; LOAD: late-onset Alzheimer’s disease; YOAD: young-onset Alzheimer’s disease; DAT: dementia of Alzheimer type; MCI: mild cognitive impairment; VAD: Vascular Dementia; DLB: dementia with Lewy bodies.

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
