# Peer review of "Exosomal miRNAs as Potential Diagnostic Biomarkers in Alzheimer’s Disease"

_pharmaceuticals, 2020, doi:10.3390/ph13090243_

Round 1
Reviewer 1 Report
The review article by Manna I, et.al. entitled “Exosomal miRNAs as potential diagnostic biomarkers in Alzheimer's disease” summarized the potential role of exosome as biomarker in Alzheimer’s detection. Starting from the characteristics to diagnosis of Alzheimer’s, authors have discussed role of miRNA in the progression of the disease. They also illustrate interestingly biomedical/tranlastional applications of exosomes. Some of the minor comments must be incorporate to improve the quality of MS.
Specific Comments are:
- Please correct the typos of line no 45- “neurotoxic amyloid…..”
- Something missing in line no 85, 251…
- Please use miRNA for MicroRNA at every place..
- Line no 137,394 etc, please write miRNA for mirnas.
- Line no 173. Citation for Burgos et al. missing..Further the line is not clear…
- A small paragraph on separation/isolation of exosome can be added (including technical limits for separation of Exosomes)
- Authors can have a refer following MS.
https://pubmed.ncbi.nlm.nih.gov/30699987/
https://pubmed.ncbi.nlm.nih.gov/29033828/
Author Response
Thank you very much for your interest in our manuscript. According to your suggestions, we made several changes and we really care that the new version of the review is now nicely improved. We have changed the abbreviations and nomenclatures as you suggested adding new references as well.
Specifically to your specific comments:
- Please correct the typos of line no 45- “neurotoxic amyloid…..”: done
- Something missing in line no 85, 251…: corrected
- Please use miRNA for MicroRNA at every place.. : done
- Line no 137,394 etc, please write miRNA for mirnas.: done
- Line no 173. Citation for Burgos et al. missing..Further the line is not clear…: corrected
- A small paragraph on separation/isolation of exosome can be added (including technical limits for separation of Exosomes): we recently published a review (in the same special issue) in which the underlined point is deeply addressed so we think that a dedicated section would be redundant (Uncovering the Exosomes Diversity: A Window of Opportunity for Tumor Progression Monitoring. Maisano D, Mimmi S, Russo R, Fioravanti A, Fiume G, Vecchio E, Nisticò N, Quinto I, Iaccino E. Pharmaceuticals (Basel). 2020 Aug 4;13(8):E180. doi: 10.3390/ph13080180).
- Authors can have a refer following MS.
https://pubmed.ncbi.nlm.nih.gov/30699987/
https://pubmed.ncbi.nlm.nih.gov/29033828/: references added
Reviewer 2 Report
This is a great comprehensive review of miRNA's and exosomes in the nervous system and AD.
Author Response
Thank you very much the reviewer for this comment.
Reviewer 3 Report
This is an excellent, accessible review which is thorough in its introduction to the topic and its discussions and conclusions. There are a small number of issues I feel need addressing, which I have split into Major and Minor:
Major.
Section 5 – this becomes quite heavy reading because of the sheer volume of miRNAs listed for each study. This can easily be summarised in a table, and the text simplified to concentrate on more significant points. For example, Lines 325 – 331 could be simplified by first simply referring to a profile of 20 miRNAs and the list in brackets put into a table. This would then allow the very interesting point about machine learning to be highlighted, and not lost amongst a list of 27 or so miRNAs. This may be in fact table 1 which is already present. Hence, information is being duplicated and can therefore be reduced in the text.
Table 1 – I am sure that the table title is not what the authors intend to have published?
Minor.
There are a number of inconsistencies in style, abbreviations and nomenclature throughout the text which need addressing. Below is a list which is not exhaustive to show what I mean. I recommend that the authors spend some time going through the text to ensure that all instances are corrected:
Line 41 – tense, should be “are” not “is”.
Line 45 (and possibly throughout) – special characters have not been reproduced in the PDF file. Check this is not the case in the final document.
Line 85 – this sentence appears to have a word missing, which means I miss the meaning of the sentence. Please check throughout the manuscript.
Line 91 – gene names need to be in italics.
Lines 107 and 108 and throughout the text – microRNA as already been defined as miRNA, please be consistent in its usage
Line 137 – mirna should be miRNA
Line 145 – spelling mistake, “exosomes”
Line 147 – RNAses, not RNases
Lines 194 and 195 – “resulting involved in the pathogenesis of AD” is grammatically incorrect and confusing. Please clarify
Line 202 – previously in the text, tau has been used without a capital. Please be consistent
Line 233 – rna should be RNA
Line 241 Alzheimer’s has already been defined as AD. Please be consistent
Line 253 – inconsistent use of amyloid-beta
Line 259 and 260 – inconsistent use of microrna
Line 278 – poor grammar – “The observed a significantly deregulation….”
Author Response
Thank you very much for your interest in our manuscript. According to your suggestions, we made several changes and we really care that the new version of the review is now nicely improved.
Specifically to your specific comments:
Major.
Section 5 – this becomes quite heavy reading because of the sheer volume of miRNAs listed for each study. This can easily be summarised in a table, and the text simplified to concentrate on more significant points. For example, Lines 325 – 331 could be simplified by first simply referring to a profile of 20 miRNAs and the list in brackets put into a table. This would then allow the very interesting point about machine learning to be highlighted, and not lost amongst a list of 27 or so miRNAs. This may be in fact table 1 which is already present. Hence, information is being duplicated and can therefore be reduced in the text. : a revision of section 5 has been made, so we hope that this version of the manuscript resulted easier to read.
Table 1 – I am sure that the table title is not what the authors intend to have published? : We apologize for the mistake in the table title. In this version of the manuscript, we have corrected the title of the table.
Minor.
There are a number of inconsistencies in style, abbreviations and nomenclature throughout the text which need addressing. Below is a list which is not exhaustive to show what I mean. I recommend that the authors spend some time going through the text to ensure that all instances are corrected:
Line 41 – tense, should be “are” not “is”. : corrected
Line 45 (and possibly throughout) – special characters have not been reproduced in the PDF file. Check this is not the case in the final document. : corrected
Line 85 – this sentence appears to have a word missing, which means I miss the meaning of the sentence. Please check throughout the manuscript. : corrected
Line 91 – gene names need to be in italics.: corrected
Lines 107 and 108 and throughout the text – microRNA as already been defined as miRNA, please be consistent in its usage: corrected
Line 137 – mirna should be miRNA: corrected
Line 145 – spelling mistake, “exosomes”: corrected
Line 147 – RNAses, not RNases: corrected
Lines 194 and 195 – “resulting involved in the pathogenesis of AD” is grammatically incorrect and confusing. Please clarify : addressed
Line 202 – previously in the text, tau has been used without a capital. Please be consistent : addressed
Line 233 – rna should be RNA : corrected
Line 241 Alzheimer’s has already been defined as AD. Please be consistent : corrected
Line 253 – inconsistent use of amyloid-beta : corrected
Line 259 and 260 – inconsistent use of microrna : corrected
Line 278 – poor grammar – “The observed a significantly deregulation….” : the grammar of the entire review has been revised
Reviewer 4 Report
The review by Manna and colleagues is yet another paper aiming at reviewing the current literature on exosomal miRNA as potential diagnostic biomarkers in Alzheimer’s disease. As it stands, the manuscript lacks direction and it failed to address the aim of the review.
While the subject is probably worthwhile writing a review, this manuscript does not reach the standard for publication. The general feeling is that it fails at bringing a fresh point of view of the use of exosomal miRNA as potential diagnostic biomarkers in AD. The review does not identify research gaps in the field, nor does it critically review the recent literature on the topic. In its present form, the review is just a list of papers that have investigated exosomal miRNA in the context of AD.
The use of poor grammar throughout the manuscript as well as very wordy and lengthy sections detract a bit the reader from the major findings. The authors are strongly encouraged to have their manuscript proofread before re-submitting it.
The first sections on the “general characteristics” and “clinical features and diagnosis” have been extensively reviewed in many other articles and should not be included in the manuscript.
Overall, the manuscript would benefit from a major reorganization to make it easier to follow. The ‘biogenesis and function’ section (2.1.) has also been covered in other review articles and can be significantly shortened by replacing part of the first long paragraph by a schematic showing how miRNA are synthesized.
Some reference numbers are shifted, which is very annoying to this reviewer and makes it hard to follow. It seems that the authors did not take the time to carefully check the reference numbers, which is a sign of sloppiness and is clearly negatively affecting the quality of the review.
Alzheimer’s disease is usually referred to as a neurodegenerative disease, and not a neurocognitive disease. The authors should include appropriate citations if they want to keep their definition of AD.
The second sentence of the Abstract can pick readers’ interest but unfortunately, the authors have not included any follow up to that statement in the text itself. A number of reviews have recently covered this topic (e.g. PMID: 27505094), yet the authors have not included them in their citations.
Lines 106-107 and many other lines: ‘deregulation’ or ‘dysregulation’? Which one is it?
Line 132-133: please rephrase.
Line 146: reference 40 is not appropriate. Please amend.
If only 10% of the circulating miRNA are contained in exosomes, how do these populations compare to the free-circulation miRNA populations which represent 90% of the miRNA?
Lines 153 and following: it is unclear to which population of miRNA the authors are referring. Exosomal or free-circulating miRNA?
Ref 51 on line 180 is not appropriate.
Ref 53 is Burgos and not Sheinerman as written in the text. The authors should have carefully checked the references before submitting their manuscript.
The topic of Section 4.1 has also been extensively reviewed in other papers and could be synthetized into a schematic.
Line 260: ref 75 is not appropriate. Please amend.
Lines 257-258: The authors need to add data (from their lab or from the literature) demonstrating that exosomal miRNA are indeed easily accessible from biological fluids. They should also consider adding some methodological information describing what is considered the best way to collect exosomal miRNA from biological fluids as this would be very relevant to readers.
The authors should consider adding subsections to Section 5, perhaps adding a subsection about CSF, one about plasma and another one about serum. At present, it is written as a big chunk of text, which is not very inviting to read. The same goes for Table 1, which could also be better organized (e.g CSF, plasma, etc. or disease type). To make it more relevant for the reader, the authors could include a column briefly describing the isolation method used to collect the exosomes from biological samples.
Line 283: it is unclear in which matrix were these 14 miRNA? Whole CSF or exosomal CSF?
Overall, Section 5 provides a list of papers that have investigated exosomal miRNA as potential biomarkers for AD. The authors do not provide any critical and/or analytical thinking as to why they picked these papers specifically. For example, one could question why in the 10 or so studies the authors cite in Table 1, there is no overlap between the different miRNA found dysregulated across the studies. That appears to be a major drawback of exosomal miRNA studies. Further, while identifying miRNA X or Y as being dysregulated could be of interest, the lack of knowledge about which function these particular miRNA support make the information somewhat less useful.
Line 377: To what kind of biochemical analyzes are the authors referring?
Line 390: the authors need to include references to support this statement (“easily accessible”). “Abundantly”? on line 144, you wrote the exosomal miRNA represent only 10% of the miRNA pool…so which one is it?
Lines 395-397: Has this been actually done (i.e., identifying the temporal changes)? Including and discussing references on that topic would be interesting to the readers.
Line 410: same comment as above about the apparent contradiction between the “strong enrichment” described here by the authors and the exosomal miRNA representing only 10% of the miRNA pool.
Line 404-405: how do you access exosomal miRNA from CNS since the authors wrote that working with CSF is hampered by technical difficulties?
The Section 7 does not bring much. The authors should refrain from using statement such as “well demonstrated” when their review paper fails at doing so.
Author Response
Thank you very much for your interest in our manuscript and for all your suggestions.
-
- We appreciated the suggestion about first section on the general characteristics and clinical features, and the second section on the biogenesis and function of miRNAs. We apologize but we have reduced only the part relating to the biogenesis of the miRNAs, as we think that section 1 can be included in the manuscript.
- We agree on the definition of Alzheimer's disease, a condition that leads to progressive brain damage and neurodegeneration, and we apologize for the mistake.
- Following her/his suggestion, regard to “the second sentence of the Abstract”, we have added details about the risk factors related to the pathogenesis of AD. We have also added the suggested references.
- We sincerely thank the reviewer for this comment. The published literature on circulating miRNAS is extremely wide, nevertheless it is still suggestive especially in relation to the AD pathogenesis. Following her/his suggestion, a revision of section 5 has been made. We have added a comment in the end part of the Section 5, and we have inserted in table 1 a column that briefly describes the methods of isolation of exosomes from the various biological fluids, used in the different works cited.
- Following her/his suggestion, a revision of section 7 has been made.
- In response to your specific suggestions:
Lines 257-258: The authors need to add data (from their lab or from the literature) demonstrating that exosomal miRNA are indeed easily accessible from biological fluids. They should also consider adding some methodological information describing what is considered the best way to collect exosomal miRNA from biological fluids as this would be very relevant to readers.
Answer: numerous articles report that exosomal mirnas are easily accessible, in fact they can be easily collected following sampling (venous or spinal tap) and subsequently isolated by means of the different techniques developed. Therefore, we have added bibliographic references that provide more information regarding the methods for the collection of exosomes from biological fluids.
Reviewer 5 Report
This Review sums up the studies on the use of exosome-contained microRNA as biomarkers for Alzheimer’s disease and other neurodegenerative conditions. In general, the Review is nicely organized and is of relevance.
Minor points
-Page 2, Line 51: since Aß has been defined in a previous paragraph, better use Aß instead of ß-amyloid peptide.
- Page 2, Line 85: missing the “ß” for Aß
- Page 2, Line 91: It is stated that mutations in the genes for APP, presenilins or ApoE are causative of early-onset AD. Is there any mutation in ApoE that cause a dominant form of AD?
- Page 4, Line 145: spelling mistake: “exososms”
Author Response
Thank you very much for the interest in our manuscript and for all your suggestions.
- We sincerely thank the reviewer for this comment. In response to your specific suggestions:
Lane 51, 85, and 145:
Answer: we have corrected the errors, according to your suggestion.
Line 91: It is stated that mutations in the genes for APP, presenilins or ApoE are causative of early-onset AD. Is there any mutation in ApoE that cause a dominant form of AD?
Answer: in this version of the manuscript we have tried to explain the above statement better.
Round 2
Reviewer 3 Report
Thank you for your efforts in addressing my points. The changes you have made makes the manuscript much easier to read.
Author Response
Thank you for all your support in the revision process.
Reviewer 4 Report
The revisions made by the authors fail by a lot to address my initial comments. Most sections of the revised manuscript are still quite lengthy and wordy and would benefit from being proofread.
The review does not bring much to the field, it fails at providing a fresh, comprehensive review of the potential of exosomal miRNA to be used as biomarkers in AD. The authors paint an idealistic picture which is far from the reality and could be misleading to readers with no or little experience with exosomes and/or miRNA.
- The authors have made some changes to their initial submission. However, many of my comments on the first version of the manuscript have not been addressed.
- As it stands, the authors failed to provide a complete and reasoned response to my comments, and I recommend rejecting their manuscript.
- Line 105: what do you mean by “familiarity”? Did you mean “familial history”?
- My comment on the second sentence of the Abstract “There are several risk factors related to the pathogenesis of AD, among these exosomes and microRNAs (miRNAs) are of particular importance” has prompted the authors to add a list of potential risk factors in the Introduction, but they have yet to provide support for the second part of the sentence which is the part that will interest readers!
- The authors also said they have reduced the part (section 2) related to the biogenesis of the miRNA but they have not. It is still my opinion that the review would clearly benefit from the addition of a schematic to help the reader follow the different steps in the biogenesis of miRNA. The authors did not comment on this suggestion made in round 1 of review. Other papers (e.g., doi.org/10.1016/j.gpb.2012.06.004; doi: 10.3389/fendo.2018.00402; https://doi.org/10.1038/nrm3838) have provided better description of miRNA biogenesis and the authors should consider shortening this section and include a schematic.
- The authors have removed a crucial piece of information I had pointed out in the first round of review (“If only 10% of the circulating miRNA are contained in exosomes, how do these populations compare to the free-circulation miRNA populations which represent 90% of the miRNA?”, “the authors need to include references to support this statement (“easily accessible”). “Abundantly”? on line 144, you wrote the exosomal miRNA represent only 10% of the miRNA pool…so which one is it?”). It is rather unclear why the authors decided to take away this statement but their decision should be justified.
- The authors have made some changes to Section 5. But, as it stands, this section do not provide any critical and/or analytical thinking which is what readers expect from a review paper. Providing a list of papers talking about AD and exosomal miRNA does not make for a good review paper. The authors have not commented on the fact (comment from first round of revision) that out of the 10 or so studies cited in Table 1, there is no overlap between the different miRNA found dysregulated across the studies. That appears to be a major drawback of exosomal miRNA studies. Further, while identifying miRNA X or Y as being dysregulated could be of interest, the lack of knowledge about which function these particular miRNA support make the information somewhat less useful.
- Replace ‘Young onset’ by ‘early onset’ in table 1
- Please note I haven’t checked the list of references
Author Response
Thank you very much for the interest in our manuscript and for all your suggestions.
Point 1.
The authors have made some changes to their initial submission. However, many of my comments on the first version of the manuscript have not been addressed.
Answer: we apologize for the lack of some answers regarding the changes made. This is due to an error we made when entering point-by-point the answers in the box. Below we provide a complete answer to the various suggestions.
Point 3.
Lane 105: what do you mean by “familiarity”? Did you mean “familial history”?
Answer: we intended to refer to the presence of the disease in the family, so we have corrected the term “familiarity” with “familial inheritance” in the text.
Point 4.
My comment on the second sentence of the Abstract “There are several risk factors related to the pathogenesis of AD, among these exosomes and microRNAs (miRNAs) are of particular importance” has prompted the authors to add a list of potential risk factors in the Introduction, but they have yet to provide support for the second part of the sentence which is the part that will interest readers!
Answer: following her/his suggestion, a revision in the end part of the Introduction has been made. Briefly, we have added some details about miRNAs as epigenetic risk factors. We think we have provided data to support this claim in subsequent sections of the manuscript, particularly sections 3, 4 and 5.
Point 5.
The authors also said they have reduced the part (section 2) related to the biogenesis of the miRNA but they have not. It is still my opinion that the review would clearly benefit from the addition of a schematic to help the reader follow the different steps in the biogenesis of miRNA. The authors did not comment on this suggestion made in round 1 of review. Other papers (e.g., doi.org/10.1016/j.gpb.2012.06.004; doi: 10.3389/fendo.2018.00402; https://doi.org/10.1038/nrm3838) have provided better description of miRNA biogenesis and the authors should consider shortening this section and include a schematic.
Answer: The part of section 2 relating to the biogenesis of miRNAs, having already reduced it, we consider it quite synthetic. In this version of the manuscript, following her/his suggestion, we have included a figure describing the main steps of this process.
Point 6.
The authors have removed a crucial piece of information I had pointed out in the first round of review (“If only 10% of the circulating miRNA are contained in exosomes, how do these populations compare to the free-circulation miRNA populations which represent 90% of the miRNA?”, “the authors need to include references to support this statement (“easily accessible”). “Abundantly”? on line 144, you wrote the exosomal miRNA represent only 10% of the miRNA pool…so which one is it?”). It is rather unclear why the authors decided to take away this statement but their decision should be justified.
Answer: unfortunately, the answer to point 6, due to our mistake, was not correctly upload in reviews report (round 1). So, our comment below.
We apologize for the contradiction on the amount of miRNAs contained in exosomes. The contradiction is linked to the two different works, on this subject, present in the literature. In a first work by Arroyo et al. (2011), an array of qRT-PCR miRNA profiles was used to measure the expression of 375 miRNA in plasma and serum fractions, obtained by ultracentrifugation, and size-exclusion chromatography. From this study it emerges that only a small amount of circulating microRNAs (about 10%) is cofractioned with the vesicles, such as exosomes. In a more recent paper Cheng et al. (2014) used NGS to profile miRNAs in various blood components and identify differences in profiles within peripheral blood versus plasma, serum, and exosomes. Furthermore, the study compared the amount of miRNA obtained from exosomes isolated via ultracentrifugation and commercial exosome isolation kits. Exosomes were found to be enriched in miRNA, in particular serum exosome contained the highest percentage of miRNA (42-48%) compared to plasma (30%). From a more careful reading of these works, we considered it appropriate to eliminate the reference to 10% in the manuscript, because it does not conform to the work of Cheng et al., which defines exosomes as a resource enriched with miRNAs.
We apologize for the mistake, but the term "abundantly" did not refer to exosomal miRNAs, but to exosomes. In this version of the manuscript we have corrected the error.
Point 7.
The authors have made some changes to Section 5. But, as it stands, this section do not provide any critical and/or analytical thinking which is what readers expect from a review paper. Providing a list of papers talking about AD and exosomal miRNA does not make for a good review paper. The authors have not commented on the fact (comment from first round of revision) that out of the 10 or so studies cited in Table 1, there is no overlap between the different miRNA found dysregulated across the studies. That appears to be a major drawback of exosomal miRNA studies. Further, while identifying miRNA X or Y as being dysregulated could be of interest, the lack of knowledge about which function these particular miRNA support make the information somewhat less useful.
Answer: in this version of the manuscript, following her/his suggestion, we have reorganized Section 5, highlighting the limitations of these studies and underlining the reason for the lack of overlap between the dysregulated miRNAs, and mentioned at the importance of their functional role.
Point 8.
Replace ‘Young onset’ by ‘early onset’ in table 1
Answer: we agree that definition “early onset” is more appropriate, but citing the work of Mckeever et al. [88], we used the term "young onset" as they themselves do.
Point 9.
Please note I haven’t checked the list of references
Answer: in this regard, already in the previous submission we had noticed that some reference numbers are shifted, as her/his suggested, and we had made all the corrections. In this last submission we have carefully looked at all references.